

# Thermal structure of the mesopause region during the WADIS-2 rocket campaign

Raimund Wörl[1], Boris Strelnikov[1], Timo P. Viehl[1], Josef Höffner[1], Pierre-Dominique Pautet[2], Michael J. Taylor[2], and Franz-Josef Lübken[1]

[1]Leibniz Institute of Atmospheric Physics, Kühlungsborn, Germany
[2]Center for Atmospheric and Space Sciences, Utah State University, Logan, Utah, United States

**Correspondence:** Raimund Wörl (woerl@iap-kborn.de)

**Abstract.** This paper presents simultaneous temperature measurements by three independent instruments during the WADIS-2 rocket campaign in northern Norway (69° N, 14° E) on 5 March 2015. Vertical profiles measured in-situ with the CONE instrument and continuous mobile IAP Fe lidar (Fe lidar) measurements during a period of 24 h, as well as horizontal resolved temperature maps of the Utah State University (USU) Advanced Mesospheric Temperature Mapper (AMTM) in the mesopause region are analysed. Vertical and horizontal temperature profiles by all three instruments are in good agreement. An harmonic analysis of the Fe lidar measurements shows the presence of waves with periods of 24 h, 12 h, 8 h, and 6 h. Strong waves with amplitudes of up to 10 K at 8 h and 6 h are found. The 24 h and 12 h components play only a minor role during these observations. In contrast only few short periodic gravity waves are found. Horizontally resolved temperatures measured with the AMTM in the OH layer are used to connect the vertical temperature profiles. In the field of view of 200 x 160 $km^2$ only small deviations from the horizontal mean of the order of 5 K are found. Therefore only weak gravity wave signatures occurred. This suggests horizontal structures of more than 200 km. A comparison of Fe lidar, rocket-borne measurements, and AMTM temperatures indicate an OH centroid altitude of about 85 km.

## 1 Introduction

The MLT (mesosphere and lower thermosphere) region is one of the key regions for the interaction of planetary waves, tides, and gravity waves. Thermal tides are typically excited by solar heating of water vapor in the troposphere, ozone in the stratosphere and mesopause region, and oxygen above 90 km altitude. They can also be excited in the troposphere by latent-heat release due to deep convection (e.g., Chapman and Lindzen, 1970; Forbes, 1984; Hagan and Forbes, 2002). Due to the excitation processes, tides have periods of the solar day (24 h) and its harmonics (12 h, 8 h, ...). Gravity waves are mostly generated in the troposphere and lower stratosphere by the flow above orographic structures, convective instabilities, wind shears, jet streams, or wave–wave interactions (e.g., Fritts and Alexander, 2003). Their propagation depends on the background wind field and its modulation by tides and planetary waves (e.g., Eckermann and Marks, 1996; Senf and Achatz, 2011). The different atmospheric layers are coupled by the transport of momentum and energy on a wide range of scales due to the propagation





and interaction of these waves. Gravity waves and tides are therefore a key driving mechanism for atmospheric processes and play an important role in their understanding.

Our knowledge of properties of the MLT region is still very limited. The main reason for this lack of knowledge is the difficulty of experimental research at this altitude. Detailed and continuous measurements are still rare (e.g., Smith, 2012).

In recent decades different techniques have been developed to investigate the MLT region. While satellites provide a global overview of the atmosphere, they do not allow to investigate variability on short time scales since they typically need several weeks to cover 24 h of local time. In-situ observations with sounding rockets allow local measurements with high resolution and precision but sporadically only. Remote sensing methods typically rely on specific phenomena which appear in the mesopause region, e.g., meteors evaporating at these altitudes. This creates layers of metallic atoms such as iron (Fe) which can be probed

by resonance lidars to derive temperatures. Furthermore, the specific chemistry of the mesopause region creates a persistent hydroxyl (OH) layer. The airglow resulting from excited OH molecules can be detected from ground based imagers to derive temperatures and horizontal resolved wave informations.

This paper shows results from experimental investigation of temperatures in the mesopause region in the frame of the WADIS-2 sounding rocket campaign, which allows to study the MLT region with high temporal and spatial resolution. The

name WADIS stands for 'Wave propagation and dissipation in the middle atmosphere: Energy budget and distribution of trace constituents'. The main goal of the campaign led by the Leibniz Institute of Atmospheric Physics (IAP), was to study propagation of gravity waves from their sources in the troposphere to their level of dissipation in the MLT and quantification of their contribution to the energy budget of the upper atmosphere. For an overview of the WADIS project and its main mission the reader is referred to Strelnikov et al. (2017). In section 2 three instruments providing MLT temperature observations with

some important parameters are described. The observations and their analysis are described in section 3. Finally, the results are discussed in section 4, and a short summary is given in section 5.

## 2  Instruments

Three instruments provided direct and indirect temperature observations in the MLT region during the WADIS-2 campaign are analysed in this study: the CONE instrument on-board the WADIS-2 rocket, the mobile IAP Fe lidar, and the Utah State

University Advanced Mesospheric Temperature Mapper. These instruments are briefly introduced in the following.

The WADIS-2 payload was equipped with two identical CONE instruments (COmbined sensor for Neutrals and Electrons) on the front and rear deck of the payload. They measure turbulence, neutral air temperature and density, and electron density with very high spatial resolution of the order of centimetres (Giebeler et al., 1993; Strelnikov et al., 2013). Data acquisition is performed independently for each sensor. As the rocket probes the upper atmosphere, the centre of mass of the payload follows

a ballistic curve but the orientation remains roughly upright. This means that the same ends of the payload are always facing upwards and downwards. The data of the respective CONE sensors pointing in the direction of the flight (front bay for the upleg, rear bay for the downleg) are analysed in this study. More details on the CONE instrument and the complete payload instrumentation can be found in Giebeler et al. (1993) and in Strelnikov et al. (2013, 2017).



Two further instruments providing ground-based observations of temperatures in the mesopause region are located at the ALOMAR observatory (Arctic Lidar Observatory for Middle Atmosphere Research), at a distance of approximately 2 km to the South of the WADIS-2 rocket launch site at the Andøya Space Center.

The mobile IAP Fe lidar (Fe lidar) has been operating at the ALOMAR observatory since summer 2014. It determines meso-
spheric temperatures and Fe densities by probing the Doppler-broadened Fe resonance line at 386 nm with a frequency-doubled alexandrite ring laser. The system is capable of observations in full daylight and allows nearly background free measurements during night and day. The receiving telescope is pointed vertically. The altitude range of accurate resonance lidar temperature measurements is limited to about 75 km to 100 km due to the extent of the meteoric Fe layer (e.g., Lautenbach and Höffner, 2004; Viehl et al., 2016). Further information about the instrument has been published by e.g., Lautenbach and Höffner (2004)
and Höffner and Lautenbach (2009).

The Utah State University (USU) Advanced Mesospheric Temperature Mapper (AMTM) (Pautet et al., 2014) was installed at the ALOMAR observatory in 2010 and measures OH (3,1) rotational temperatures at the altitude of the OH layer. An altitude range of 82 km to 90 km was found for the OH centroid height (e.g., von Zahn et al., 1987; Baker and Stair, 1988). The AMTM allows temperature observations with high temporal and spatial (horizontal) resolution during the night even in
presence of auroras. During the day, the solar background rises above the OH emission which prevents further observations. An intensity and temperature map with a resolution of 320 x 256 pixels (0.6 x 0.6 $km^2$ per pixel) is taken from the OH layer every 30 s. This corresponds to an overall area of 200 x 160 $km^2$ centred at the observation site (ALOMAR). In Pautet et al. (2014) more information on the instrument and its development are given.

The horizontal arrangement of the measurement volumes at the OH layer altitude is shown in Fig. 1. The horizontally
resolved AMTM temperature maps allow to connect the vertical datasets, which are point measurements in the horizontal domain. As illustrated in Fig. 1, the WADIS-2 rocket was launched roughly in north-western direction. The marks labelled 'rocket upleg' and 'rocket downleg' indicate the positions at which the rocket passed through the OH layer during the up- and the downward flight paths, respectively. The 'Fe lidar' mark defines the location of Fe lidar measurements at the ALOMAR observatory, where the AMTM is also located. The crosses on the map mark only the location, not the size of the CONE and
the lidar measurement volumes. Their horizontal extent is of the order of several cm for the CONE sensors and several 10 m for the Fe lidar.

## 3   Observations

The WADIS-2 rocket was launched from the Andøya Space Center in northern Norway (69° N, 14° E) on 5 March 2015 at 01:44 UT. Good weather conditions with clear sky in the period from 4 March 2015, 10:00 UT to 5 March 2015, 12:00 UT
allowed for simultaneous and nearly continuous Fe lidar and AMTM measurements with high data quality. No relevant ground based optical measurements were obtained during the days before or after the launch due to poor weather conditions. In the following we focus on the discussion of data obtained within a 24 h period around the launch night, i.e. from 4 March 2014 12:00 UT to 5 March 2015 12:00 UT.



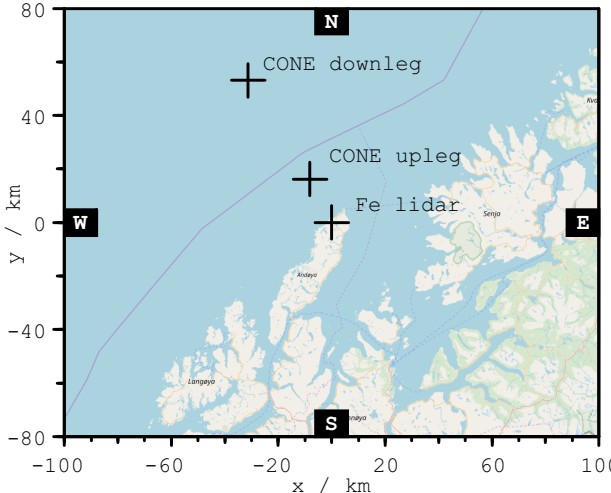

**Figure 1.** Map showing the area covered by the AMTM at the centroid altitude of the OH layer at about 86 km. The points of measurement at that altitude are marked for each instrument (see text for further details).

## 3.1 WADIS-2 rocket (CONE)

Figure 2 shows the vertical temperature profiles obtained from the CONE measurements on the rocket payload. The horizontal distance of the up- and downleg measurements is around 50 km at an altitude of 86 km (see Fig. 1). The duration of the flight through the mesopause altitude range between 70 km and 110 km is about 50 s both for the up- and downleg. In between the two profiles, the rocket spent about 2 min in the apogee range above 110 km. An effective altitude resolution of the temperature measurements with the CONE sensor is about 200 m (Rapp et al., 2003). The shapes of both profiles are very similar above 80 km and differ only in small details. The uncertainty is about 2 K at 70 km and increases with altitutde (up to about 5 K at 110 km) (Strelnikov et al., 2013). For the most part the differences are within the uncertainty. The temperature shows two maxima at about 80 km and 100 km. Such features are often referred to as "mesospheric inversion layer" (e.g., Hauchecorne et al., 1987; Hauchecorne and Maillard, 1992; Meriwether and Gerrard, 2004). The most prominent deviations between the profiles are of the order of 10 K and occur at around 95 km and below 80 km.

## 3.2 Fe lidar

For direct comparison with the CONE data, the temperature profile obtained by the Fe lidar at the time of the rocket launch is shown in Fig. 2. In contrast to the in situ measurements, the Fe lidar data is integrated over 60 min and 1 km in altitude (with 0.2 km intervals) to derive a temperature profile. The centre of the averaging window is 01:45 UT. As shown in Fig. 1 for an altitude of 86 km, the distance of the Fe lidar profile to the upleg of the rocket is about 10 km. Therefore, the maximum distance between all three profiles is around 60 km. The temperature profile measured with the Fe lidar is in good agreement with the profiles measured with the CONE sensors on the rocket.



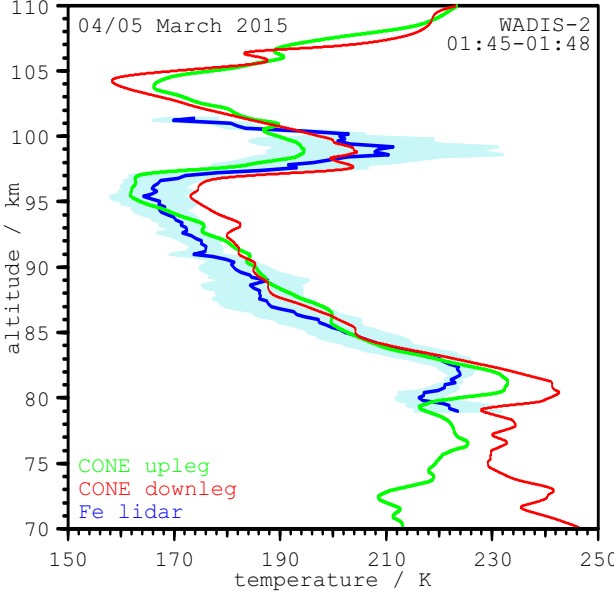

**Figure 2.** Vertical temperature profiles at the launch time of the WADIS-2 rocket. The red and green curves show the in-situ measurements with the rocket payload (CONE). The upleg (green) and the downleg (red) profiles are measured with two independent instruments at the ends of the payload. The Fe lidar (blue) profile is integrated over 60 min and centred at 01:45. In addition, the RMS of all Fe lidar profiles within a period of ±60 min around launch time (blue shaded area) show the variability in time.

Figure 3 shows the temporal evolution of the temperatures in the mesopause region. Temperatures are calculated in 15 min and 0.2 km intervals using running means of 60 min and 1 km width, respectively. Typical uncertainties are of the order of 2 K. At the edge of the layer where metal densities are the lowest and the backscatter signal therefore the weakest the uncertainties are not more than 10 K. The observable altitude range as well as measurement uncertainties vary over time as

5    absolute Fe densities and the vertical extent of the metal layer changes throughout the day (e.g., Höffner and Fricke-Begemann, 2005; Viehl et al., 2016). Mean temperatures during the observation period are around 190 K which is typical for the polar mesopause region in the late winter state. Strong wave like modulations with amplitudes of more than 30 K are clearly present. Lübken et al. (2011) observed similar wave-like modulations with the same instrument at the conjugate latitude in Antarctica. That study found surprisingly strong tidal signatures in temperature and Fe density observations with an harmonic analysis

10    (24 h and 12 h components) in summer. We perform a similar harmonic analysis to further investigate the apparent wave-like temperature structure in Fig. 3. The data shown in Fig. 3 is averaged in intervals of 60 min and 1 km vertical resolution. A non-linear function of sinusoidal components with fixed periods $P_i$ is then fitted to the observations according to the relation

$$T(t,z) = A_0(z) + \sum_i A_i(z) \cdot \cos\left(\frac{2\pi \cdot (t - \Phi_i(z))}{P_i}\right) \tag{1}$$



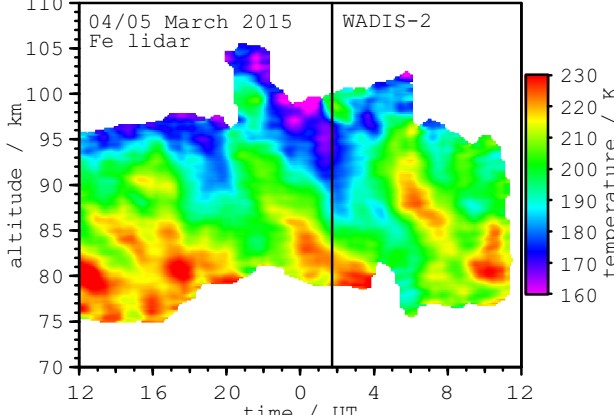

**Figure 3.** Temperatures measured with the Fe lidar in the 24 h period around the WADIS-2 launch at 01:44 UT (vertical line). Measurement uncertainties are smaller than 10 K throughout the altitude range and decrease to around 2 K towards altitude with highest Fe density.

where $A_i(z)$ are the amplitudes, $\Phi_i(z)$ the phases (at the time of the maximum amplitude), and $z$ the altitude. In addition to the periods of 24 h and 12 h investigated by Lübken et al. (2011), we also consider the higher harmonic 8 h and 6 h components. All four components are optimised simultaneously but independently for each altitude using a least square fit routine. The seasonal variation in this analysis of 24 consecutive hours can be neglected contrary to the method presented by

Lübken et al. (2011).

Figure 4 shows the result for the exemplary altitude of 86 km over time compared with the measurement. The main variation with large amplitudes is nearly fully described by the model using 4 components. The remaining variations are small compared to overall modulation which is of the order of 30 K. The mean square error of the deviation is 4.6 K. The amplitudes for the 24 h and the 12 h components are 4.8 K and 2.8 K, respectively. The higher 8 h and 6 h components show amplitudes of 6.5

K and 1.4 K.

Figure 5 shows the amplitudes and phases derived for all altitudes. Not all available temperature measurements shown in Fig. 3 are included here, as altitudes below 80 km and above 95 km are not fully covered throughout the 24 h observation period. Noteworthy is the large amplitude of the 8 h component at all altitudes which partly exceeds 10 K as well as the strong 6 h component (above 87 km). Neither the 24 h nor the 12 h components reach comparable amplitudes. Altitudes at which the

amplitudes of the harmonic fit is smaller than their uncertainty are removed in the phase plots. All 4 components show a clear phase response in altitude without large phase jumps. This suggests clear waves in the data with periods near the chosen fit periods (24 h and higher harmonics). Since every altitude is fitted independently, unclear wave structures would lead to a more incoherent phase response and amplitde. A linear fit of the phase is used to estimate the phase slope for every component: a vertical wavelength and a phase are calculated using the slope and the position of the fitted lines at 86 km. Table 1 summarises

the derived vertical wavelengths ($\lambda_{\text{vertical}}$) and the phases at 86 km. Phases are given in local solar time (LST) which was UT + 51 min at the launch site on 5 March 2015.



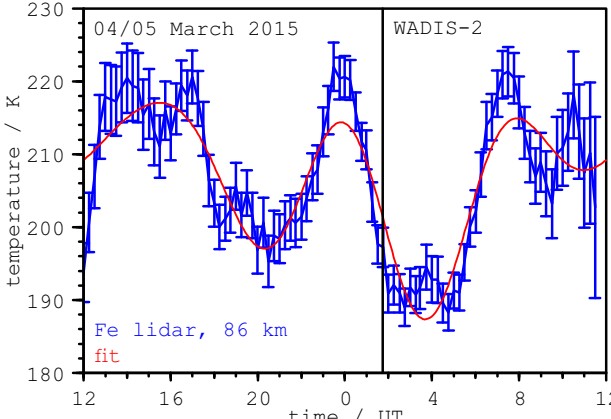

**Figure 4.** Comparison of the temperature variations measured with the Fe lidar (blue) at an altitude of 86 km with the reconstruction (magenta) using the results of the harmonic wave analysis. The amplitudes of the 24 h, 12 h, 8 h and 6 h components are 4.8 K, 2.8 K, 6.5 K and 1.4 K.

| periode | 24 h | 12 h | 8 h | 6 h |
|---|---|---|---|---|
| $\lambda_{\text{vertical}}$ | 43 km | 22 km | 23 km | 30 km |
| phase | 16 LST | 14 LST | 18 LST | 15 LST |

**Table 1.** Vertical wavelengths ($\lambda_{\text{vertical}}$) and phases (at 86 km) derived from the harmonic fit of Fe lidar temperatures.

Figure 6 shows the observed temperature deviation from the mean in comparison to the deviation of a temperature field reconstructed using only the components derived in the harmonic analysis. This reconstruction is in good agreement with the observations. The residuals do not exhibit any systematic deviations (not shown). Differences between measured and reconstructed temperatures mainly consist of small variations with periods smaller than 6 h which are not considered in the fit.

## 3.3 Advanced Mesospheric Temperature Mapper (AMTM)

The Advanced Mesospheric Temperature Mapper (AMTM) provides horizontally resolved temperature maps during night-time conditions (sun more than 9° under the horizon). In early March, first temperature maps are available at around 17:30 UT and observations end at around 05:00 UT. On the particular day of the WADIS-2 campaign, the data quality was somewhat affected by passing clouds between 22:00 UT and 24:00 UT as well as after 04:00 UT.

### 3.3.1 Horizontal temperature structure

Four representative examples of horizontal temperature maps are shown in Fig. 7. All panels show absolute temperatures (same scale with 30 K range). The boundary areas of the maps are partly shaded by other equipment on the roof or dust (the grey areas in the upper and lower part of the figures). Panels (a) and (b) are taken at 18:30 UT and 21:30 UT, that is, 7.2 h and



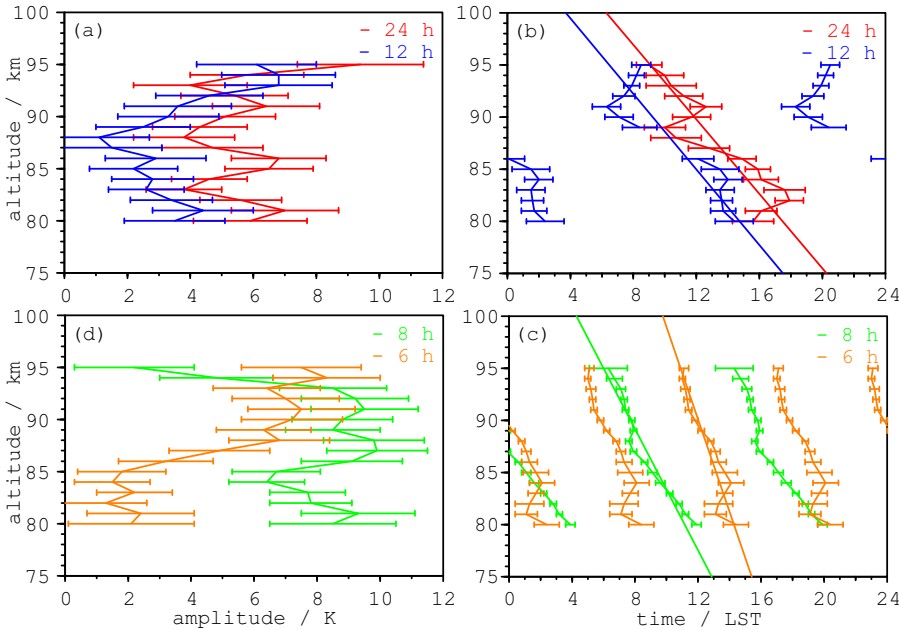

**Figure 5.** Amplitudes (a), (c) and phases (b), (d) of the 24 h (red), 12 h (blue), 8 h (green), and 6 h (orange) wave components during the 24 h period around the WADIS-2 launch. The phase shift with altitude is approximated with straight lines and used to calculate vertical wavelengths. The approximation corresponds to vertical wavelength of 43 km for the 24 h, 22 km for the 12 h, 23 km for the 8 h and 30 km for the 6 h component.

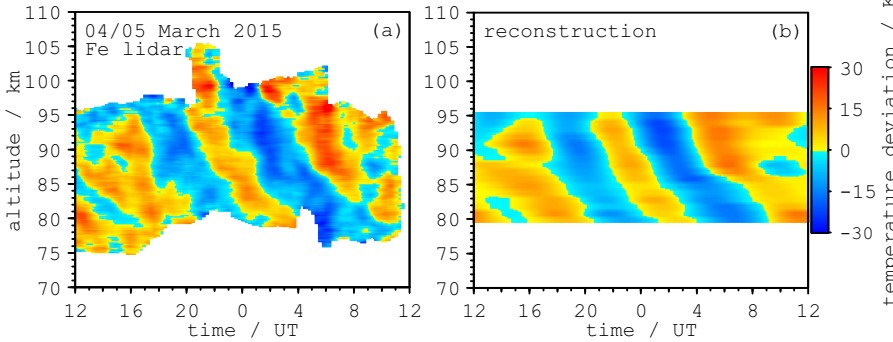

**Figure 6.** Temperature observations with the Fe lidar (a) and the reconstruction using the 4 harmonic components (b) derived in the analysis.

4.2 h before the rocket launch. Panel (c) shows the temperature structure at the time of the rocket launch at 01:44 UT. This observation was taken one minute before the launch, since the camera was overexposed due to the brightness of the rocket engine one minute later. Panel (d) shows the situation at the end of the AMTM observations at 03:30 UT (1.8 h after rocket launch). Due to higher temperatures at the beginning of the observations (see Fig. 8 or Fig. 9) the temperature map in panel



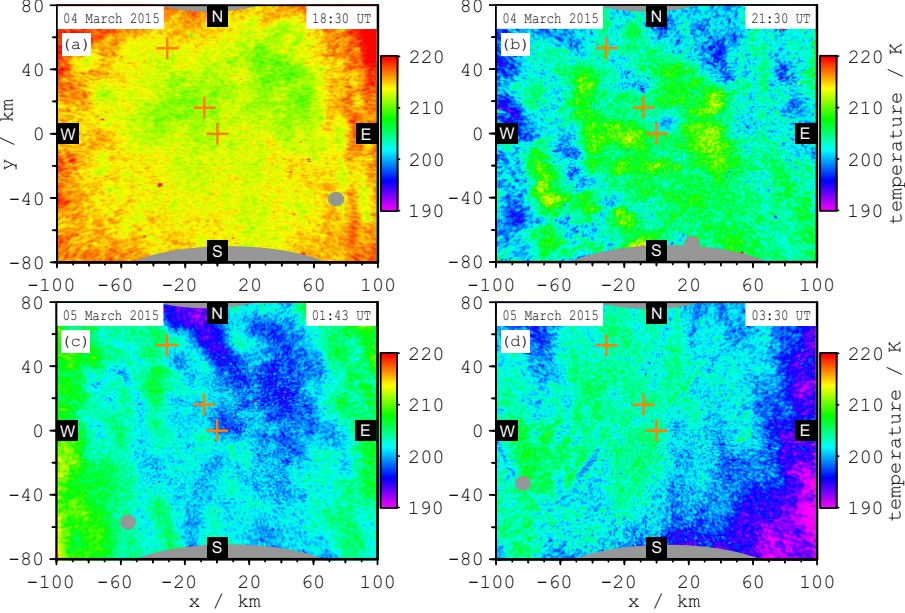

**Figure 7.** Temperature maps measured with the AMTM. In (a), (b) and (d) the typical situation during the observation time is shown. (c) shows the situation at the rocket launch time. Very little gravity wave activity is apparent, which was the typical condition during this night.

(a) (18:30 UT) differ from the other panels (more red and yellow instead of green and blue). These examples show only a slight increase or decrease at the edges and otherwise exhibit no systematic variations throughout the horizontal extent of the observations. In particular, the regions with vertical observations by CONE and Fe Lidar (indicated by orange crosses, compare to Fig. 1) show only very small temperature variations. Observations with more gravity wave activity visible in AMTM maps

are also available. Examples of common AMTM and lidar measurements with more gravity wave activity at the ALOMAR observatory can be found in e.g., Bossert et al. (2014).

   Figure 8 shows the temporal evolution of the temperatures at different locations. The selected positions are the marked observation volumes of the other instruments (sub-array of 9 x 9 pixels which corresponds to about 5 x 5 $km^2$) and show the time-dependent OH temperature evolution measured with the AMTM. The time series is averaged by a 5 min running mean

window shifted in 30 s intervals. The temperature differences between the selected locations does not vary significantly beyond the 2–3 K measurement uncertainty of the AMTM at most of the time. Temperature variations in time are dominated by long term variations of several hours. There are also waves with periods around 5 min, but with small amplitudes of only a few K (not shown). At the exact time of the launch, no temperature differences are discernible. The variations are nearly synchronous at all locations. The choice of the sub-array size (here 9 x 9 pixels) used to calculate a representative temperature for every

position has nearly no effect on the results. A single pixel as well as an area larger than 9 x 9 pixels yield to nearly the same temperatures (not significant different, not shown). This implies structures larger than the field of view (200 x 160 $km^2$) in



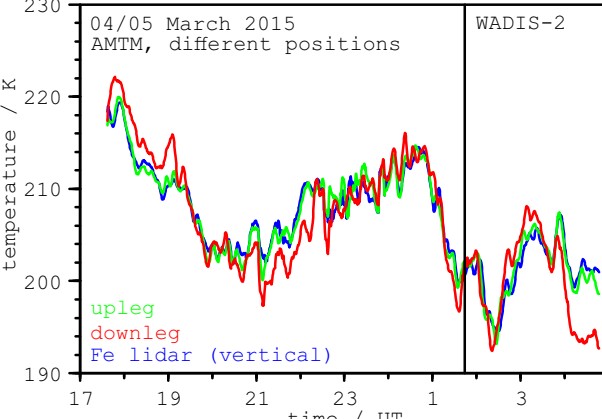

**Figure 8.** Temporal evolution of AMTM temperature observations derived from sub-arrays of $9 \times 9$ pixels (about $5 \times 5$ km$^2$) at the centre (blue), the rocket upleg (green), and the rocket downleg (red), respectively. The time series is smoothed with a Hanning filter of 5 min width.

horizontal direction, since neither the average area size (sub-array) nor the position on the map have significant influence on the temperature results.

### 3.3.2 Comparison of horizontal and vertical temperature observations

In Fig. 9, temperatures from AMTM observations are compared to vertical Fe lidar observations. An area of $9 \times 9$ pixels
around the centre of the AMTM temperature map is taken and smoothed using a 60 min running mean window. The Fe lidar temperatures shown here are averaged correspondingly. Furthermore, a vertical averaging needs to be applied to the Fe lidar data to take the altitude distribution of the OH layer and the subsequent altitude weighing of AMTM temperature measurements into account. However, the actual vertical extent of the OH layer during this observation is unknown. A Gaussian distribution with a constant FWHM of 9 km is frequently assumed (e.g., Pautet et al., 2014). This distribution is applied to the Fe lidar
data as a weighting function, simulating the AMTM's vertical averaging throughout the OH layer. The centroid altitude of the weighting function is then shifted in altitude to find the best agreement between absolute AMTM and Fe lidar temperatures. The best agreement is found at a centroid altitude of $84 \pm 1$ km, where both instruments show the same temperatures within the error bars throughout the full observation period. This is a plausible altitude of the OH layer and hereafter assumed to be the peak altitude during the night. The best temperature agreement is found at a slightly higher centroid altitude of $85 \pm 1$ km
if relative temperature variations and not absolute temperatures are considered. However, this difference of 1 km (within the altitude uncertainty) is not significant for the following discussion as the weighing function smooths the values from different altitudes at a comparatively broad FWHM of 9 km. The exact knowledge of the OH layer altitude is not important to study horizontal structures, since relative instead of absolute temperatures can be used .



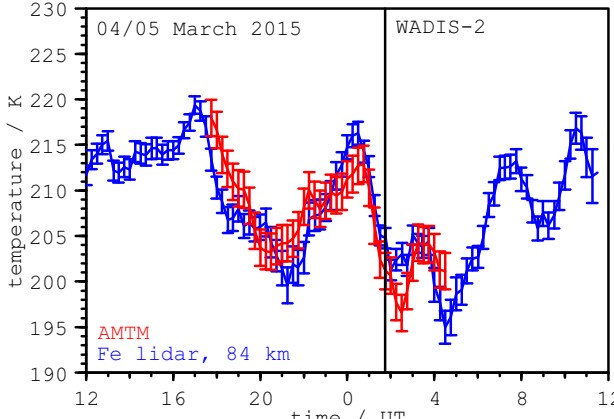

**Figure 9.** Time series of OH temperatures (red) in comparison with the Fe lidar temperatures (blue). Lidar temperatures are averaged in altitude with a Gaussian weighting of 9 km FWHM and peak altitude of $84 \pm 1$ km. Temperatures of the AMTM are taken from a sub-array of 9 x 9 pixels at the locations of the Fe lidar and smoothed by a 60 min running mean.

## 4 Discussion

The temperature profiles obtained by the CONE instrument during the up- and downleg (50 km distance) show a remarkably good agreement. Temperatures are comparatively smooth above 80 km and only a few small structures disturb the dominating large structure. Small differences of about 5 K are observed at around 87 km or at 104 km and 106 km. The uncertainty is
about 2 K at 70 km, 5 K at 100 km and 25 K at 110 km (Strelnikov et al., 2013). The CONE sensors measure the whole profile during the short flight time of about 50 s for the altitude range from 70 km to 110 km. Waves with short periods are therefore not averaged out of the temperature profiles derived from the CONE instrument. While the up- and downleg temperature profiles are in good overall agreement above 80 km, they deviate from each other at lower altitudes. Small-scale structures with vertical sizes of several km are largely absent at higher altitudes, but noticeable in the lowest range. A possible
explanation for the observed behaviour is an enhanced gravity wave activity at altitudes below 80 km. A more detailed analysis of the wave activity at lower altitudes is not possible with the given observations. Above 80 km altitude, the two profiles are also in very good agreement with the Fe lidar measurements. This is noteworthy since all three profiles were measured at different locations and are separated by a distance of up to 60 km. While the CONE sensors have a high time resolution, the Fe Lidar profile is averaged over a 60 min period. The Fe lidar's measurement uncertainty is about 3 K in that altitude range. The
good agreement (within the uncertainties) and very similar variations of the 3 profiles suggest a dynamic structure at a larger scale than the measurement distances of about 60 km.

Figure 2 shows the vertical profiles at launch time and additionally the RMS of all temperature profiles of the Fe lidar (in 15 min intervals) within $\pm 60$ min around the launch (blue shaded area). The variability of up to 20 K within only one hour is significantly larger than the observed differences between the three profiles (or two instruments). Variations on scales shorter
than the measurement distance would cause either larger differences or a phase shift between the profiles.



The temporal evolution of the thermal structure of the mesopause altitudes is obtained by analysing the full dataset of the Fe lidar. As noted above, the temperature structure is dominated by waves with comparatively long periods of 8 h and 6 h. This occasionally causes vertical temperatures profiles not following the simple model of a mesopause region with a negative temperature gradient below and a positive temperature gradient above. Such inverted temperature profiles are sometimes referred

to as 'mesospheric inversion layers' and occur quite regularly in the presence of strong long periodic waves (e.g., Meriwether and Gardner, 2000). An harmonic analysis demonstrates that the main variability is given by only 4 harmonic components (24 h, 12 h, 8 h and 6 h), adding up to more than 30 K temperature difference. In addition to these 4 components, only waves with periods smaller than 6 h remain. We note that the dataset the harmonic analysis is performed on, is not sensitive to waves with periods smaller than about 2 h due to the 60 min integration time of the Fe lidar temperatures.

Fe lidar measurements with the same instrument at the opposite latitude in the southern hemisphere at Davis, Antarctica (69° S) had revealed clear and regular 24 h (diurnal) and 12 h (semi-diurnal) tides with large amplitudes (Lübken et al., 2011). During the WADIS-2 campaign we find a dominating 8 h wave structure. An additional strong 6 h component may also be present above 87 km. In contrast to the observations in the southern hemisphere, 24 h and 12 h variations were comparatively weak. We note that this current analysis is limited to a single day (24 h) and not a composite of 180 h of measurements during

a period of 12 days as in Lübken et al. (2011). Due to the high variability in tidal phases and amplitudes (e.g., Murphy et al., 2006; Baumgarten et al., 2018) the results of a single day (our dataset) are to be expected to deviate to some extend in phase (several hours) and amplitude from an average (Lübken et al., 2011).

The presence of clear 24 h and 12 h tides as reported in Lübken et al. (2011) for Davis suggests that tides could be an explanation for the measured 24 h and 12 h waves. Nevertheless, gravity waves have to be taken into account, in particular

for the higher frequency components. It is not possible to exclude them but there are some arguments for tides we discuss below. The analysis of the phase response (see Fig. 5) allows an estimation of the vertical wavelength and the phase. A vertical wavelength of 43 km for the 24 h wave and 22 km for the 12 h wave is found in this study. The phase of the 24 h component at 86 km derived in this study has a maximum at about 16:00 LST, the 12 h compound at about 14:00 LST. The data used in Lübken et al. (2011) was obtained at different location (southern instead of northern hemisphere) and also during a different

season (summer instead of spring). Therefore, a more detailed comparison does not necessarily result in a good agreement due to the seasonal variability. Lübken et al. (2011), however, report only a slightly different vertical wavelength of 30 km for observations of 24 h temperature tides and 40 km for tides in Fe densities, as well as a phase maximum at 86 km at 13:00 LST. Model calculations for thermal tides were done for temperatures, e.g., for 24 h by Forbes (1982a) and for 12 h by Forbes (1982b) at 60° latitude during equinox conditions. The phase response reported in Forbes (1982a, b) allows to estimate the

phase and vertical wavelength for the 24 h and 12 h tide in the same way. A vertical wavelength (at about 90 km) in the range of 30–35 km (24 h tide) and 20–30 km (12 h tide) in that analysis is in good agreement with the findings in this study (40 km and 22 km). A comparison of the phase does not add up to a clear result. While the phase maximum of the 24 h tide (08:00 LST) calculated by Forbes (1982a) differs from our finding (16:00 LST), the phase maximum of the 12 h tide (14:00 LST, Forbes (1982b)) is in good agreement with our result (also 14:00 LST). The model calculations in Forbes (1982a, b) do not

include short term variability in tides which might be the reason for the differences. The short available dataset in this study





(24 h) can also explain the differences, especially for the 24 h tide, since the dataset covers only one period to fit. Nevertheless, the good agreement of the model calculations and the WADIS-2 dataset in this study is a presumption that at least the 24 h and the 12 h waves are of tidal origin.

Winds from radar measurements and temperatures from airglow measurements at polar latitudes are discussed relating the
8 h and the 6 h tides (Younger et al., 2002; Wu et al., 2005; Smith et al., 2004). In contrast to our results the higher harmonic components (8 h and 6 h) were found to be significantly smaller than the 24 h and 12 h tides. Vertical wavelength in the range of 25 km to 45 km (spring) are reported for the 8 h tide (Younger et al., 2002; Wu et al., 2005). This is in good agreement with our finding of 30 km. 30 to 50 km for the 6 h tide as reported in Smith et al. (2004) is also in good agreement with our result of 30 km and might suggest that the strong 8 h and 6 h components found in this paper are tides. In contrast to the good agreement
in vertical wavelength the large amplitudes of the 8 h and 6 h waves are unexpected. Higher harmonic tides, in particular the 8 h component, are discussed in several, also theoretical papers (e.g., Thayaparan, 1997; Taylor et al., 1999; States and Gardner, 2000; Akmaev, 2001; Younger et al., 2002; Batista et al., 2004; Smith et al., 2004; Wu et al., 2005; Beldon et al., 2006; Jacobi and Fytterer, 2012; Lilienthal et al., 2018). Generally the amplitude of the 8 h component and higher harmonics are assumed to be significantly smaller than the 24 h and 12 h components. A small number of publications report 8 h tides with large
amplitudes (e.g., Taylor et al., 1999; Thayaparan, 1997), but at mid latitudes. They also report a 8 h tidal component with amplitudes smaller compared to the 24 h and the 12 h component in averaged datasets. However, single events (days) with 8 h tides and amplitudes larger than the 24 h and the 12 h tides are described, too.

Some good reasons can be found that not only the 24 h and the 12 h waves, but also the strong 8 h and the 6 h waves are tides, as described above. However, a larger dataset with global coverage is necessary to determine the temporal and the spatial
structure, and to separate the tidal and the gravity wave components unambiguously.

In addition to the vertical information provided by the CONE and Fe lidar instruments, horizontal information can be extracted form the AMTM temperature maps. The AMTM measurements are limited to the OH layer altitude. While this is only a single altitude or rather an altitude range of about 9 km (FWHM) due to the thickness of the OH layer, the analysis reveals interesting horizontal structures. A peak altitude of the OH layer at about 85 km and a FWHM of 9 km are plausible
values and seem to be good estimates since both absolute temperatures and also the structures in the time series show a good agreement with the Fe lidar. Without the presence of small structures (with large amplitudes), like gravity waves, the large structures should dominate the horizontal AMTM temperature maps. In Bossert et al. (2014) and Pautet et al. (2014) some examples of typical AMTM observations with gravity wave activity are described. Figure 7 shows the typical situation during the night of the WADIS-2 launch. In contrast to other examples (Bossert et al., 2014; Pautet et al., 2014) no clear
small structures are visible. The whole map shows temperature differences of no more than 10 K. This does not exclude the presence of smaller perturbations. Because temperatures are derived from the OH layer with a thickness of about 9 km, vertical structures of a similar size or smaller than the layer width cannot be resolved. A simple way to estimate the relation between datasets measured at different locations in a time series is to pick the temperatures at the locations of interest in the AMTM maps and compare them. Figure 8 shows an impressive synchronous evolution at all locations at a resolution of 5 min. Taking
the measurement uncertainty of the AMTM of about 2-3 K into account there are only a few short periods where the deviations





are significant. At around 22:00 UT and at the end at about 04:00 UT, the deviations come from disturbances by clouds. Comparing the different locations in the horizontal resolved AMTM temperature maps shows that the large structures found in all vertical profiles are not only long periodic, but have also large horizontal scales since no systematic deviation or change in temperature was observed in horizontal direction. Clearly, a dynamic variation with a horizontal extent larger than the field

of view of the AMTM, which corresponds to an area of about $200 \times 160 \ \mathrm{km}^2$, was present around the time of the WADIS-2 launch. Large structures in horizontal direction are in principle an indication of tidal structures. However, the field of view is limited to $200 \times 160 \ \mathrm{km}^2$ and it is not possible to exclude gravity waves with horizontal wavelengths of the same order.

Temperatures derived from the Doppler-broadening of metal atoms with the Fe lidar and from excited rotational OH (3,1) transitions with the AMTM are in very good agreement to each other. Both the absolute temperatures and the deviation during

the observation period suggest an OH centroid altitude between 84 km and 86 km. However, in context of this paper such statements refer only to the observation in this single night and the assumption of a fixed altitude and layer shape is not justified in all cases (e.g., Zhao et al., 2005; Dunker, 2017).

## 5   Summary

We have analysed the temperature structure in the mesopause region during the WADIS-2 rocket campaign in March 2015.

Temperatures in the night of the rocket launch were dominated by larger waves at 8 h and 6 h periods and are nearly undisturbed by gravity waves of smaller scales. The lidar measurements show waves with typical periods for 24-h and 12-h tides. A strong 8 h wave (and at higher altitudes also a 6 h wave) is dominating the variations in temperatures and might be also tides. Amplitudes of up to 10 K for this single harmonic component exceed the corresponding amplitudes of the longer 24 h and 12 h components of 6 K. Disturbances by waves or other structures with smaller periods play only a minor.

We found only unclear wave pattern with comparable small amplitudes of only a few K in the horizontal AMTM measurements. The long periodic tidal-like variations in time domain show no structures in the observable area of $160 \times 200 \ \mathrm{km}^2$. The structure size of the dynamic variation dominating this night has to be larger than 200 km, as temperatures change quasi-synchronously at all locations. As result of this situation, the rocket borne measurements show two vertical temperature profiles with very similar structures, although they were measured at a horizontal distance of 50 km.

The Fe lidar, which was located 10 km away from the WADIS-2 measurements, provides further vertical temperature profiles which show the same features as the profiles measured with CONE during the rocket flight. Especially the altitude range around the OH layer at about 85 km show the same structures and absolute temperatures.

In this case the OH temperatures show a remarkably good agreement with the lidar temperatures during the whole night, if we assuming an OH density centroid altitude of 85 km. Below 80 km stronger temperature deviation and also small variations

are the result of increasing influence of small scale gravity waves.

*Data availability.*   The data are available upon request from the corresponding author.





*Competing interests.* The authors declare that they have no conflict of interest.

*Acknowledgements.* This work was supported by the German Space Agency (DLR) under grant 50OE1001 (project WADIS). The design and initial development of the AMTM was supported under the AFOSR DURIP grant F49620-02-1-0258. Its installation and operations at ALOMAR were supported under the NSF collaborative grant AGS-1042227. This project was also partly supported by the Deutsche
5   Forschungsgemeinschaft (DFG, German Research Foundation) under project LU1174/8-1 (PACOG), FOR1898 (MS-GWaves). The work was also partly supported by the Bundesministerium für Bildung und Forschung (BMBF, Federal Ministry of Education and Research) under project D/553/67210010 (ROMIC-GWLcycle).



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
