# Peer review of "Thermal structure of the mesopause region during the WADIS-2 rocket campaign"

_Atmospheric Chemistry and Physics, 2018_

## Referee Comment (RC1) · Anonymous Referee #1 · 7 Sep 2018

A review on the paper "Thermal structure of the mesopause region during the WADIS-2 rocket campaign" by Raimund Wörl, Boris Strelnikov, Timo P. Viehl, Josef Höffner, Pierre-Dominique Pautet, Michael J. Taylor, and Franz-Josef Lübken

Although the paper provides no new knowledge about the temperature regime of the mesopause, solar tides and gravity waves, the paper is interesting in terms of a comparison of simultaneous temperature measurements by three independent instruments (the rocket CONE instrument, the IAP Fe lidar and the AMTM mapper).

I recommend the paper for publication after minor revisions.

Minor comments:

P. 6, L.17: "unclear wave structures" It is better to replace it with "random wave structures".

P. 6, L. 18-19: "A linear fit of the phase is used to estimate the phase slope for every component: a vertical wavelength and a phase are calculated using the slope and the position of the fitted lines at 86 km."

This is the most problematic issue. I do not understand this technique of estimating a vertical wavelength, having just a 15 km altitude range of the data profiles shown in Fig. 5. Some strong assumptions are required. This technique should be clarified in more detail. Besides, uncertainties for the estimated vertical wavelengths and phases should be presented in Table 1.

P. 8, Capture to Fig. 5: "Figure 5. Amplitudes (a), (c) and phases (b), (d) . . ." Labels (c) and (d) should be swapped around.

P. 12, L. 27-29: "A vertical wavelength (at about 90 km) in the range of 30–35 km (24 h tide) and 20–30 km (12 h tide) in that analysis is in good agreement with the findings in this study (40 km and 22 km)."

First. In Table 1, a vertical wavelength of 43 km for the 24 h wave is shown (not 40 km). Second. Having vertical wavelength of 30-35 km (from Forbes, 1982a) and the found vertical wavelength of 43 km, one cannot say that "it is in good agreement". This sentence should be corrected/rephrased.

P. 13, L. 1-2: "Winds from radar measurements and temperatures from airglow measurements at polar latitudes are discussed relating the 8 h and the 6 h tides (Younger et al., 2002; Wu et al., 2005; K. Smith et al., 2004)."

It is worth indicating here that Dalin et al. (2017) have demonstrated significant solar tidal components (24h, 12 and 8h) both in the PMSE strength and wind velocity components in the polar mesopause.

P. 13, L. 4-5: "This is in good agreement with our finding of 30 km." In Table 1, a vertical wavelength of 23 km for the 8 h wave is shown. This value should be corrected.

P. 14, L. 7-9: ". . .suggest an OH centroid altitude between 84 km and 86 km. However, in context of this paper such statements refer only to the observation in this single night and the assumption of a fixed altitude and layer shape is not justified in all cases (e.g., Zhao et al., 2005; Dunker, 2017)."

It is worth adding other important references of this important topic on the variability of OH layer characteristics in height and time: Perminov et al., 1999; Melo et al., 2000; Liu and Shepherd, 2006; Khomich et al., 2008; Grygalashvyly et al., 2014.

Additional references:

Dalin, P., S. Kirkwood, N. Pertsev, V. and Perminov (2017). Influence of solar and lunar tides on the mesopause region as observed in polar mesosphere summer echoes characteristics. Journal Geophysical Research-Atmospheres, 122. https://doi.org/10.1002/2017JD026509.

Grygalashvyly, M., G.R. Sonnemann, F.-J. Lübken, P. Hartogh, U. Berger (2014). Hydroxyl layer: mean state and trends at midlatitudes. J. Geophys. Res. Atmos. 119, 12391–12419. http://dx.doi.org/10.1002/2014JD022094.

Khomich, V.Yu., A.I. Semenov, N.N. Shefov (2008). Airglow as an indicator of upper atmospheric structure and dynamics. Springer-Verlag, Berlin, Heidelberg. http://dx.doi.org/10.1007/978-3-540-75833-4.

Liu, G.,and G.G. Shepherd (2006). An empirical model for the altitude of the OH nightglow emission. Geophys. Res. Lett. 33, L09805. http://dx.doi.org/10.1029/2005GL025297.

Melo, S.M.L., R.P. Lowe, and J.P. Russell (2000), Double-peaked hydroxyl airglow profiles observed from WINDII/UARS, J. Geophys. Res., 105, D10, 12,397-12,403.

---

## Referee Comment (RC2) · Anonymous Referee #2 · 28 Sep 2018

The study presents temperature measurements taken by three instruments during a 24 hour period. Thus it can be classified as a case study.

The combination of AMTM, lidar, and CONE measurements is unique, but simultaneous observations taken by AMTM and lidar as well as lidar and CONE instruments have been published before. Given that Worl et al. show that the addition of the CONE data to AMTM and lidar data provides mostly redundant information concerning large scale gravity waves or tides, I think this manuscript does not provide enough new results for an ACP publication. Therefore, I suggest that the authors focus more on quantitative analysis of gravity waves and tides, and in particular small-scale perturbations. I recommend a major revision or re-submission for this manuscript.

Major comment 1

[Figure]

According to the authors, the IAP Fe lidar has been operating at the ALOMAR observatory since summer 2014 and the AMT since 2010. It is probably reasonable to assume that a large amount of data was collected by both instruments in the following years until the rocket launch. It is surprising to me that the authors did not even try to classify conditions observed during the rocket launch with respect to the climatological mean state or at least typical conditions. Instead, the authors spend significant time speculating about tides, where most of the speculation is based on a comparison with measurements taken in the southern hemisphere at the wrong time of the year. Discussion of gravity waves is limited to the statement "In contrast to other examples (Bossert et al., 2014; Pautet et al. 2014) no clear small structures are visible". The authors make no attempt whatsoever to quantify gravity waves in their observations. Thus, the manuscript is merely a presentation of measurement data without any meaningful analysis. Conclusions drawn by the authors are weak.

According to the Review Criteria https://www.atmospheric-chemistry-and-physics.net/peer_review/review_criteria.html reviewers are asked to answer the question "Does the manuscript represent a substantial contribution to scientific progress within the scope of Atmospheric Chemistry and Physics (substantial new concepts, ideas, methods, or data)?" Yes, the data are new in the sense that every observation of the atmosphere is different. However, temperature measurements in the mesopause region are hardly anything new and numerous case studies were published during the last 30 years. Thus, without thorough analysis, publishing the data is of low scientific significance, and the manuscript in its current state might be seen as an attempt to boost the publication statistics with minimum effort.

I am not arguing that the data should not be published. On the contrary, the observational data presented in this study has potential. But the authors should invest the time and analyze the data, critically review their hypothesis, and draw meaningful conclusions.

Suggestions

1) The speculation about tides can be resolved with the help of meteor radar data (a meteor radar is located in the vicinity of the launch site). Retrieving tidal components and phases from meteor winds is common practice.

2) Keograms created from AMTM data can provide information on the horizontal structure of the larger-scale waves and direction of propagation.

3) There is a paper by Hildebrand et. al discussing winds and temperatures above ALOMAR (https://doi.org/10.5194/acp-17-13345-2017). This work could be a starting point for gravity wave analysis.

4) Include temperature data taken by the co-located Rayleigh lidar (I assume it was running during the WADIS-2 campaign). Extending the altitude range down to ∼70 km may help to distinguish between tides and gravity waves.

Major comment 2

The authors do not meet the data policy (https://www.atmospheric-chemistry-and-physics.net/about/data_policy.html) which clearly states the request for depositing the data in reliable (public) data repositories.

Minor comments

Page 2, line 4: What does "nearly background free measurements" mean?

Page 3, line 20: What is the idea behind "connecting vertical data sets"? What does that mean in practice?

Page 4, line 15: What is the typical temperature error of these lidar measurements?

Caption of Figure 2: "the RMS of all Fe lidar profiles within a period of +/- 60 min around launch time" – The integration time is 60 min for all profiles, right? Are you saying that you computed the RMS of all profiles which have their centers in the interval 60 min before launch to 60 min after launch? How many profiles did you use?

Page 5, line 3: I suggest you move the information concerning resolutions and errors to the beginning of this section, before you discuss the lidar temperature profile in Figure 1.

Page 5, line 6: "Mean temperatures during the observation period are around 190 K which is typical for the mesopause region..." - Well, that depends on what altitude you are talking about. According to your Figure 3, the mesopause is at the top of your profiles or above.

Page 5, line 4: "not more than 10 K" – Did you limit the vertical extent of your temperature profiles to altitudes where the error is <10 K?

Page 7, line 1: "deviation from the mean in comparison to the deviation of a temperature field reconstructed..." – I assume you computed the mean for each altitude and removed it?

Page 10, line 6: "are averaged correspondingly" - What temperatures are averaged? Earlier you stated that the integration time is 60 min. Please clarify.

Page 11, lines 15-16: "very similar variations of the 3 profiles suggest a dynamic structure at a larger scale than the measurement distances of about 60 km" – That statement is not well supported by your data. In my opinion, all you can safely say here is that there appears to be no significant variability at horizontal scales below about 60 km.

Page 11, lines 19-20: "Variations on scales shorter than the measurement distance would cause either larger differences or a phase shift between the profiles" – Please clarify. What are you referring to? Are you comparing the three profiles which were taken approximately at the same time, or are you referring to the temporal evolution of the lidar measurements? In any case, a phase shift between the profiles causes larger RMS differences, unless the phase shift is 2Pi.

Page 12, lines 30-33: I do not think you can say that your value (43 km according to Table 1, or 40 km as written in the text?) is in good agreement with a vertical wavelength

of 30-35 km reported by Forbes.

Page 13, line 8: According to your Table 1 the vertical wavelength of the 8-hour component is 23 km and not 30 km. Please make the values consistent.

Page 14, lines 15-16: "were dominated by larger waves . . . and are nearly undisturbed by gravity waves of smaller scales" – This statement is sort of trivial. It is clear that small-scale waves, in particular waves with small vertical wavelengths, become quickly unstable as amplitudes grow. Therefore, amplitudes of small-scale waves are in general smaller than larger scale waves, and the larger scale waves appear to be undisturbed by the small-scale waves. A more interesting question is whether amplitudes are close to the saturation limit. See for example Smith et al., Evidence for a saturated spectrum of atmospheric gravity waves, 1987.

Typos, grammar, wording

Page 2, line 14: "allows us to study" or "allows for studies"

Page 2, line 23: something is wrong with the grammar

Page 2, line 3: "allows to" is ungrammatical, there are several instances in the text

Page 6, line 8: mean square error -> mean squared error

Page 6, line16: phase response -> phase progression?

Page 9, line 1: differ from -> is different from?

Page 10, line 7: altitude distribution of the OH layer -> vertical profile of the OH layer?

Page 10, line 12: is found at a centroid altitude -> is found for the centroid altitude 84. . .

Page 10, line 17: is not important to study horizontal structures -> is not important for studies of horizontal structures?

Caption of Figure 9: at the locations of -> at the location of

Page 11, line 2: Temperatures cannot be smooth -> The profiles are…

Page 11, line 9: lowest range -> lower part?

Page 12, line 1: thermal structure of the mesopause altitudes -> thermal structure in the mesopause region?

Page 12, line 5: long periodic waves -> waves with long periods

Page 14, line 19: "play only a minor." – sentence incomplete

---

## Author Comment (AC1) · 19 Nov 2018

A review on the paper "Thermal structure of the mesopause region during the WADIS2 rocket campaign" by Raimund Wörl, Boris Strelnikov, Timo P. Viehl, Josef Höffner, Pierre-Dominique Pautet, Michael J. Taylor, and Franz-Josef Lübken Although the paper provides no new knowledge about the temperature regime of the mesopause, solar tides and gravity waves, the paper is interesting in terms of a comparison of simultaneous temperature measurements by three independent instruments (the rocket CONE instrument, the IAP Fe lidar and the AMTM mapper).

I recommend the paper for publication after minor revisions.

Minor comments:

P. 6, L.17: "unclear wave structures" It is better to replace it with "random wave structures".

changed

P. 6, L. 18-19: "A linear fit of the phase is used to estimate the phase slope for every component: a vertical wavelength and a phase are calculated using the slope and the position of the fitted lines at 86 km."

This is the most problematic issue. I do not understand this technique of estimating a vertical wavelength, having just a 15 km altitude range of the data profiles shown in Fig. 5. Some strong assumptions are required. This technique should be clarified in more detail. Besides, uncertainties for the estimated vertical wavelengths and phases should be presented in Table 1.

page 7, line 31: a short explanation is added

P. 8, Capture to Fig. 5: "Figure 5. Amplitudes (a), (c) and phases (b), (d) …" Labels (c) and (d) should be swapped around.

changed

P. 12, L. 27-29: "A vertical wavelength (at about 90 km) in the range of 30–35 km (24 h tide) and 20–30 km (12 h tide) in that analysis is in good agreement with the findings in this study (40 km and 22 km)."

First. In Table 1, a vertical wavelength of 43 km for the 24 h wave is shown (not 40 km). Second. Having vertical wavelength of 30-35 km (from Forbes, 1982a) and the found vertical wavelength of 43 km, one cannot say that "it is in good agreement". This sentence should be corrected/rephrased.

numbers are updated in the table and the text

page 13, line 31 ff. and page 14: text is corrected and rephrased

P. 13, L. 1-2: "Winds from radar measurements and temperatures from airglow measurements at polar latitudes are discussed relating the 8 h and the 6 h tides (Younger et al., 2002; Wu et al., 2005; K. Smith et al., 2004)."

It is worth indicating here that Dalin et al. (2017) have demonstrated significant solar tidal components (24h, 12 and 8h) both in the PMSE strength and wind velocity components in the polar mesopause.

Page 14, line 20: Dalin et al. (2017) is mentioned

P.13,L.4-5: "This is in good agreement with our finding of 30 km."

In Table 1, a vertical wavelength of 23 km for the 8 h wave is shown. This value should be corrected.

numbers are corrected

P. 14, L. 7-9: "…suggest an OH centroid altitude between 84 km and 86 km. However, in context of this paper such statements refer only to the observation in this single night and the assumption of a fixed altitude and layer shape is not justified in all cases (e.g., Zhao et al., 2005; Dunker, 2017)."

It is worth adding other important references of this important topic on the variability of OH layer characteristics in height and time: Perminov et al., 1999; Melo et al., 2000; Liu and Shepherd, 2006; Khomich et al., 2008; Grygalashvyly et al., 2014.

Additional references

 Dalin, P., S. Kirkwood, N. Pertsev, V. and Perminov (2017). Influence of solar and lunar tides on the mesopause region as observed in polar mesosphere summer echoes characteristics. Journal Geophysical Research-Atmospheres, 122. https://doi.org/10.1002/2017JD026509.

Grygalashvyly, M., G.R. Sonnemann, F.-J. Lübken, P. Hartogh, U. Berger (2014). Hydroxyl layer: mean state and trends at midlatitudes. J. Geophys. Res. Atmos. 119, 12391–12419. http://dx.doi.org/10.1002/2014JD022094.

Khomich, V.Yu., A.I. Semenov, N.N. Shefov (2008). Airglow as an indicator of upper atmospheric structure and dynamics. Springer-Verlag, Berlin, Heidelberg. http://dx.doi.org/10.1007/978-3-540-75833-4.

Liu, G.,and G.G. Shepherd (2006). An empirical model for the altitude of the OH nightglow emission. Geophys. Res. Lett. 33, L09805. http://dx.doi.org/10.1029/2005GL025297.

Melo, S.M.L., R.P. Lowe, and J.P. Russell (2000), Double-peaked hydroxyl airglow profiles observed from WINDII/UARS, J. Geophys. Res., 105, D10, 12,397-12,403.

page 15, line 31: additional references are mentioned

---

## Author Comment (AC2) · 19 Nov 2018

The study presents temperature measurements taken by three instruments during a 24 hour period. Thus it can be classified as a case study. The combination of AMTM, lidar, and CONE measurements is unique, but simultaneous observations taken by AMTM and lidar as well as lidar and CONE instruments have been published before. Given that Worl et al. show that the addition of the CONE data to AMTM and lidar data provides mostly redundant information concerning large scale gravity waves or tides, I think this manuscript does not provide enough new results for an ACP publication. Therefore, I suggest that the authors focus more on quantitative analysis of gravity waves and tides, and in particular small-scale perturbations. I recommend a major revision or re-submission for this manuscript.

Major comment 1

According to the authors, the IAP Fe lidar has been operating at the ALOMAR observatory since summer 2014 and the AMT since 2010. It is probably reasonable to assume that a large amount of data was collected by both instruments in the following years until the rocket launch. It is surprising to me that the authors did not even try to classify conditions observed during the rocket launch with respect to the climatological mean state or at least typical conditions. Instead, the authors spend significant time speculating about tides, where most of the speculation is based on a comparison with measurements taken in the southern hemisphere at the wrong time of the year. Discussion of gravity waves is limited to the statement "In contrast to other examples (Bossert et al., 2014; Pautet et al. 2014) no clear small structures are visible". The authors make no attempt whatsoever to quantify gravity waves in their observations. Thus, the manuscript is merely a presentation of measurement data without any meaningful analysis. Conclusions drawn by the authors are weak.

According to the Review Criteria https://www.atmospheric-chemistry-andphysics.net/peer_review/review_criteria.html reviewers are asked to answer the question "Does the manuscript represent a substantial contribution to scientific progress within the scope of Atmospheric Chemistry and Physics (substantial new concepts, ideas, methods, or data)?" Yes, the data are new in the sense that every observation of the atmosphere is different. However,

temperature measurements in the mesopause region are hardly anything new and numerous case studies were published during the last 30 years. Thus, without thorough analysis, publishing the data is of low scientific significance, and the manuscript in its current state might be seen as an attempt to boost the publication statistics with minimum effort.

I am not arguing that the data should not be published. On the contrary, the observational data presented in this study has potential. But the authors should invest the time and analyze the data, critically review their hypothesis, and draw meaningful conclusions.

The main goal of the paper is giving an overview about the mesopause temperature structure during the WADIS-2 campaign as background information for further studies. Small-scale variations of gravity waves and turbulence are part of a separate paper and out of the scope of this paper. A recent submitted paper deals e.g. with small scale variations gravity waves and turbulence:

Strelnikov, B. et. al.: Simultaneous in situ measurements of small-scale structures in neutral, plasma, and atomic oxygen densities during WADIS sounding rocket project. Atmos. Chem. Phys. Discussion, 2018, submitted.

Gravity waves are often discussed in literature by selecting periods of strong gravity wave activities as mentioned above, focusing on short duration measurements of a few hours, but not covering the required measurement length for identifying tides/long period waves. Because of this selection process the importance of tides/long period waves are ignored and most of the time not even discussed. We note that most measurements in literature (lidar and OH-imager) do not even have the capability of true 24-hour measurements or measurements are too short because of weather or other limitations. The references given (Pautet et al., 2014; Bossert et al., 2014) are examples of this selected view to the mesosphere.

In contrast, we show in this paper for the day of the WADIS-2 campaign that small-scale gravity waves can be absent for most of a day in the field of view of 200x160 km$^2$, playing nearly no role (see Fig 8). Fig. 9 shows for three locations of lidar and rocket the agreement of the AMTM-temperature measurements for different locations as an example for all measurements within the field of view of the AMTM. For the whole field of view this is already discussed in the manuscript. We note that a comparison of different instruments at different locations with high accuracy is not possible in the presence of small-scale gravity waves. This is to the best of our knowledge the first data set allowing such a detail comparison in literature and therefore new. The agreement of all instruments in absolute temperatures cannot be expected and is often showing a time dependence over the measurement period. This is clearly not the case here. We note that 3 different methods of temperature observations are compared here.

As an example that this in general is not the case is shown at the lower altitude part in figure 2. The temperatures measured with CONE on the rocket are the same at altitudes above 80 km. In the range 70 km to 80 km the profiles look less smoothed due to small scale gravity waves and there is a temperature

difference of more than 30 K (also mentioned in the text). We note that this measurements are only two minutes and 50 km apart. In this unique case such major differences do not occur throughout the day at the altitude of the AMTM measurement in a field of view of 200x160 km$^2$ which is an unexpected finding not published elsewhere and in agreement with the lidar observations.

The conditions during the WADIS-2 campaign therefore allow to our knowledge for the first time a detailed comparison of several instruments at different locations with high accuracy

The Fe lidar can provide information on long periodic variations due to the possibility for solar background free daylight measurements. This optimisation reduces the sensitivity for short periodic waves. The short period, small-scale gravity are not observed because of the very small amplitudes of these waves during this particular day.

There are not many measurements during the winter season with the lidar. It is usually not operated for short good weather periods of only a few hours, which is the typical situation during the winter in Andoya. As already mentioned in the text, there are not enough measurements in February or March to give a more detailed overview. As mention below, a fitted temperature climatology is used to compare with the daily mean temperature. A climatology of tides or waves with long periods are hardly possible. A lot of measurements don't cover a full day or more, due to weather conditions and other operation limits.

The AMTM can measure only during night-time and therefore can hardly cover a full day, which is needed to get reliable tidal information. Measurements are only available during and around the winter season.

If it is assumed (reviewer comment to saturated gravity wave amplitudes) the small scale waves have such small amplitudes because of saturation effects it is necessary to include tidal amplitudes (see also comment to saturated gravity waves below) to characterise the small scale waves. Without tidal information for the winter season as background temperature field (not enough measurements in this season with the Fe lidar) a classification in context of a gravity wave climatology can hardly be done.

Suggestions

1) The speculation about tides can be resolved with the help of meteor radar data (a meteor radar is located in the vicinity of the launch site). Retrieving tidal components and phases from meteor winds is common practice.

This study is dealing with temperatures. Since there is no simple and direct connection between wind and temperature variations (e.g. different modes for wind and temperature), we don't think it's helpful to add wind information. A tidal analysis of the winds provides different results as for the temperatures. With one single day of measurements it is not possible to understand the differences in wind and temperatures and we think at the end there are more

new questions than answers. The comparison of temperatures and wind should be done with a larger dataset and in a separate study.

2) Keograms created from AMTM data can provide information on the horizontal structure of the larger-scale waves and direction of propagation.

As mentioned in the text, the variations with long periods are practical simultaneous on the whole AMTM temperature maps. The large scale variation (red to blue in the keograms) don't show a clear propagation direction (19:00 maybe from west, 22:00 maybe from east?).

[Figure]

It is correct the Keograms show some small scale wave (mainly 5 min period with small amplitudes). But from time to time there were clouds or fog during this night. There are several time slots which are disturbed by the effects of the clouds or fog, e.g.: around 21:00, 23:00, 03:00, 04:00.

The Keogram is already submitted by Strelnikov et al. as mentioned above

4) Include temperature data taken by the co-located Rayleigh lidar (I assume it was running during the WADIS-2 campaign). Extending the altitude range down to~70 km may help to distinguish between tides and gravity waves.

It is not possible to extend the altitude range with the RMR lidar in this way. Similar to the lower part of the Fe lidar data set (75 - 80 km) the RMR lidar doesn't provide temperatures during the whole 24-hour in the altitude range 70 - 80 km due to daylight conditions. With only partial coverage no meaningful results can be expected above 70 km altitude in terms of long period waves/tides. For this reason we show e.g. no results from the wave analysis (figure 6) for the altitude range below 80 km. The lower altitude (< 70 km) are out of the scope of this paper and shown by Strelnikov et al.

Major comment 2

The authors do not meet the data policy (https://www.atmospheric-chemistry-andphysics.net/about/data_policy.html) which clearly states the request for depositing the data in reliable (public) data repositories.

page 16, line 20: data is available now, at the mentioned repository.

Page 2, line 4: What does "nearly background free measurements" mean?

Page 3, line 8: free of solar background is meant, text is changed

Page 3, line 20: What is the idea behind "connecting vertical data sets"? What does that mean in practice?

page 3, line 21: text is changed to clarify, that the horizontal resolved variations ca help to decide if the structures of two, or several vertical profiles belong to the same phenomena or are two independent features.

Page 4, line 15: What is the typical temperature error of these lidar measurements?

page 5, line 5: typical temperature error is mentioned now at the beginning of the paragraph

Caption of Figure2: "the RMS of all Fe lidar profiles with in a period of +/- 60 min around launch time" – The integration time is 60 min for all profiles, right? Are you saying that you computed the RMS of all profiles which have their centers in the interval 60 min before launch to 60 min after launch? How many profiles did you use?

Caption figure 2: correct, mentioned now in the caption

page 13, line 3: also mentioned in the text

Page5, line 3: I suggest you move the information concerning resolutions and errors to the beginning of this section, before you discuss the lidar temperature profile in Figure 1.

page 5, line 5: done

Page 5, line 6: "Mean temperatures during the observation period are around 190 K which is typical for the mesopause region…" - Well, that depends on what altitude you are talking about. According to your Figure 3, the mesopause is at the top of your profiles or above.

Page 6, line 3: mean temperature profile of the day is compared with a profile of the climatology, figure 4 added.

Page 5, line 4: "not more than 10 K" – Did you limit the vertical extent of your temperature profiles to altitudes where the error is <10 K?

page 5, line 6: Correct, temperatures with uncertainties larger than 10 K are not shown, written in the text now.

Page 7, line 1: "deviation from the mean in comparison to the deviation of a temperature field reconstructed…" – I assume you computed the mean for each altitude and removed it?

page 8, line 3: correct, sentence changed

Page 10, line 6: "are averaged correspondingly" - What temperatures are averaged? Earlier you stated that the integration time is 60 min. Please clarify.

page 8, line 11: yes, integration time is already 60 min for the lidar profiles. Only the AMTM temperatures are averaged, to get a similar time resolution. Sentence changed

Page 11, lines 15-16: "very similar variations of the 3 profiles suggest a dynamic structure at a larger scale than the measurement distances of about 60 km"– Thatstatement is not well supported by your data. In my opinion, all you can safely say here is that there appears to be no significant variability at horizontal scales below about 60 km.

page 12, line 17: modified, large structures are mentioned as a possibility, not as fact, now.

Page 11, lines 19-20: "Variations on scales shorter than the measurement distance would cause either larger differences or a phase shift between the profiles" – Please clarify. What are you referring to? Are you comparing the three profiles which were taken approximately at the same time, or are you referring to the temporal evolution of the lidar measurements? In any case, a phase shift between the profiles causes larger RMS differences, unless the phase shift is 2Pi.

page 13, line 3: paragraph is rewritten to make it clearer.

Page 12, lines 30-33: I do not think you can say that your value (43 km according to Table1,or 40 km as written in the text?) is in good agreement with a vertical wavelength of 30-35 km reported by Forbes.

page 14, line 9 ff.: numbers are updated and corrected, the text is changed

Page 13, line 8: According to your Table 1 the vertical wavelength of the 8-hour component is 23 km and not 30 km. Please make the values consistent.

all numbers has been checked and corrected, there was something wrong.

Page 14, lines 15-16: "were dominated by larger waves … and are nearly undisturbed by gravity waves of smaller scales" – This statement is sort of trivial. It is clear that small-scale waves, in particular waves with small vertical wavelengths, become quickly unstable as amplitudes grow. Therefore, amplitudes of small-scale waves are in general smaller than larger scale waves, and the larger scale waves appear to be undisturbed by the small-scale waves. A more interesting question is whether amplitudes are close to the saturation limit. See for example Smith et al., Evidence for a saturated spectrum of atmospheric gravity waves, 1987.

page 16, line 4: sentences are changed to "play only a minor role", instead of disturbed.

Correct, small scale wave are limited at smaller amplitudes than waves with large scales. Nevertheless, there are situations (usually the cases which are selected for gravity wave analysis in other studies) with larger wave amplitudes (mentioned example in the text). We have a situation without significant waves with periods from 6 h to 5 min and only minor important small scale/period waves (5 min).

The limit of amplitudes is always function of the amplitudes of all waves. As mentioned above we don't have enough common and long lidar and AMTM measurements to classify the conditions in a way of a climatology including tides and gravity waves.

Typos, grammar, wording

Page 2, line 14: "allows us to study" or "allows for studies"

corrected

Page 2, line 23: something is wrong with the grammar

corrected

Page 2, line 3: "allows to" is ungrammatical, there are several instances in the text

sentences changed

Page 6, line 8: mean square error -> mean squared error

corrected

Page 6, line16: phase response -> phase progression?

changed

Page 9, line 1: differ from -> is different from?

changed

Page 10, line 7: altitude distribution of the OH layer -> vertical profile of the OH layer?

changed

Page10, line12: is found at a centroid altitude->is found for the centroid altitude 84…

changed

Page 10, line 17: is not important to study horizontal structures -> is not important for studies of horizontal structures?

changed

Caption of Figure 9: at the locations of -> at the location of

corrected

Page 11, line 2: Temperatures cannot be smooth -> The profiles are…

corrected

Page 11, line 9: lowest range -> lower part? Page 12, line 1: thermal structure of the mesopause altitudes -> thermal structure in the mesopause region?

changed

Page 12, line 5: long periodic waves -> waves with long periods

changed

Page 14, line 19: "play only a minor." – sentence incomplete

corrected